# Regional Control and Optimization of Heat Input during CMT by Wire Arc Additive Manufacturing: Modeling and Microstructure Effects

**DOI:** 10.3390/ma14051061

**Published:** 2021-02-24

**Authors:** Furong Chen, Yihang Yang, Hualong Feng

**Affiliations:** School of Materials Science and Engineering, Inner Mongolia University of Technology, Hohhot 010051, China; cfr7075@163.com (F.C.); a15598049404@163.com (H.F.)

**Keywords:** wire arc additive manufacturing, cold metal transfer, aluminum-magnesium alloy, EBSD, microstructure

## Abstract

Wire arc additive manufacturing (WAAM) of aluminum-magnesium (Al–Mg) ER5356 alloy deposits is accomplished by cold metal transfer (CMT). During the process, the temperature change of the alloy deposits has a great influence on molding quality, and the microstructure and properties of alloy deposits are also affected by the complex thermal history of the additive manufacturing process. Here, we used an inter-layer cooling process and controlled the heat input process to attempt to reduce the influence of thermal history on alloy deposits during the additive process. The results showed that inter-layer cooling can optimize the molding quality of alloy deposits, but with the disadvantages of a long test time and slow deposition rate. A simple and uniform reduction of heat input makes the molding quality worse, but controlling the heat input by regions can optimize the molding quality of the alloy deposits. The thermophysical properties of Al-Mg alloy deposits were measured, and we found that the specific heat capacity and thermal diffusivity of alloy deposits were not obviously affected by the temperature. The microstructure and morphology of the deposited specimens were observed and analyzed by microscope and electron back-scatter diffraction (EBSD). The process of controlled heat input results in a higher deposition rate, less side-wall roughness, minimum average grain size, and less coarse recrystallization. In addition, different thermal histories lead to different texture types in the inter-layer cooling process. Finally, a controlled heat input process yields the highest average microhardness of the deposited specimen, and the fluctuation range is small. We expect that the process of controlling heat input by model height region will be widely used in the WAAM field.

## 1. Introduction

With the development of modern industry, metal-based additive manufacturing has been widely surveyed by international researchers [1,2,3]. Different methods of metal-based additive manufacturing have been proposed and studied, such as direct metal laser sintering (DMLS), selective laser melting (SLM), and electron beam melting (EBM). Wire arc additive manufacturing (WAAM) with the arc as the heat source has the advantages of fast formation, low cost, and simple equipment [4,5]. Compared with using laser or electron beams as the heat source, arcs have higher energy efficiency [6].

Arcs generated by tungsten inert gas (TIG), metal inert gas (MIG), or plasma arc welding (PAW) are used as the heat source, and the welding wire is used as filler material [7]. In various WAAM processes, cold metal transfer (CMT) technology provides a faster cooling rate and almost no splash deposition, leading to better molding quality of the deposited specimen [8,9,10]. For aluminum alloys with lightweight structure, it is a better choice to use CMT for WAAM. However, during the continuous deposition process, the complex thermal history usually results in poor quality of the deposition specimens [11,12]. Various processes that improve the molding quality of the deposited specimens also lead to changes in the microstructure and properties.

Many researchers have proposed methods to optimize the molding quality of alloy deposits. Liu et al. [13] found that CMT is suitable for additive manufacturing of 4043 aluminum alloy. The parameters of the deposited specimens were investigated, and the reciprocating scanning method was suitable for CMT-deposited specimens. Although the problem of collapse at both ends of the deposited specimen was solved, there were defects caused by temperature, namely a narrow bottom and wide top. The ideal shape has a uniform width. Lehmann et al. [14] found a correlation between the heat accumulated during deposition and the geometric changes in the reference component, and proposed using in-situ temperature monitoring methods to achieve optimal component quality. The method of inter-layer cooling with in-situ temperature monitoring can optimize the molding quality of alloy deposits, but it requires more equipment, slow deposition rate, and more complex operation.

In the present study, we propose that the heat input can be controlled by varying the temperature of the specimen during deposition. Our goal was to improve the molding quality of alloy deposits, simplify the process, and increase the deposition rate. CMT was used for WAAM of aluminum-magnesium (Al–Mg) ER5356 alloy deposits. By measuring the thermophysical properties of the Al-Mg alloy deposits, we were able to analyze changes in the temperature field during deposition, and then design the process parameters to control the change of heat input in different regions. At the same time, we used the inter-layer cooling process and compared the morphology, microstructure, and microhardness of specimens created by these different processes. An idea to optimize the wire arc additive manufacturing of aluminum alloy wires was provided.

## 2. Materials and Methods

Aluminum-magnesium (Al-Mg) ER5356 alloy welding wire (Elektriska Svetsnings AktieBolaget, Shanghai, China) with a diameter of 1.2 mm was used in this experiment. This is a kind of universal welding material with wide application, which is suitable for welding and wire arc additive manufacturing of aluminum alloy. 6061 aluminum alloy (SW Aluminum, Chongqing, China) substrate with 4 mm thickness was selected for the test. The composition of the wire and substrate are detailed in Table 1.

The initial process parameters are provided in Table 2. The wire feed speed was automatically matched with the welding current. The CMT-WAAM system used in the test is shown in Figure 1, and was adapted from the conventional CMT welding system. The CMT-WAAM system included a TPS2700 CMT (Fronius, Pettenbach, Austria) arc welding power supply, a self-developed 3D slide table, and a programmable logic controller (PLC) console for the slide table.

As shown in Figure 2, the wall with 150 mm length and 18 layers was stacked in the form of reciprocating stacking. The specimen was cut along the cross section by wire cut electrical discharge machining (WEDM). A series of thermal analyses were carried out with a Netzsch LFA 457 laser thermal conductivity instrument (Netzsch, Selb, Germany). Two K-type NiCr-NiSi thermocouples were used as a temperature sensor to measure temperature changes during the deposition process. The specimen microstructure was observed with a metallographic microscope (Zeiss, Oberkochen, Germany), and electron back-scatter diffraction (EBSD) analysis was carried out with a FEI-QUANTA 650 scanning electron microscope (FEI, Hillsborough, FL, USA). The microhardness of different areas of the specimen was measured with a Vickers hardness tester (Matsuzawa, Akita, Japan).

## 3. Results and Discussion

### 3.1. Temperature Field of WAAM

During the WAAM process, the heat transmission and distribution of the material are fairly regular. The specific heat capacity (C), thermal diffusion coefficient (α), thermal conductivity (λ), and density (ρ) of the material determine the heat transfer rate inside the specimen, as well as the temperature values of different regions in the specimen under thermal diffusion. The thermal diffusion coefficient of a material is a measure of the rate at which the change of temperature at one point in the object is transferred to another point. Figure 3a shows the thermal diffusion coefficient of a Al-Mg alloy deposited specimen. The thermal diffusion coefficient of the material increased with the increase of temperature, but the fluctuation of the thermal diffusion coefficient was not large. The range of the value was only 42.69 to 49.08 mm^2^/s. Because the thermal diffusivity of the specimen was less affected by the temperature change, the change of thermal diffusivity had no obvious effect on internal heat conduction.

The specific heat capacity (C) of a material indicates the ability of the material to absorb the heat required to reach a certain temperature, i.e. the amount of heat absorbed (or released) per unit of mass per temperature unit. Figure 3b shows the measured specific heat capacity curve of Al-Mg alloy deposited specimens with the corresponding temperatures. The specific heat capacity of the material changed slightly and fluctuated between 0.89 and 1.03 J·g^−1^·K^−1^. From the Equation of thermal diffusivity
(1)$$\alpha =\frac{\lambda }{\rho C}$$
the specific heat capacity (C) and thermal diffusivity (α) of a material tend to be constant and the change of material density (ρ) is small; thus its thermal conductivity (λ) will fluctuate within a small range. The thermal conductivity is the heat transferred through 1 square meter within a certain time. During the deposition process, the thermal conductivity inside a specimen is much larger than that on the surface of the specimen and in the surrounding air. Assuming that no heat transfer takes place between the specimen and the air during welding, the Fourier Equation [15] can be used:(2)$$q=-\;\lambda \frac{\partial T}{\partial x}$$
where q is the heat flux per unit area, and ∂T/∂x is the temperature gradient in the x direction. When applied to WAAM, q becomes the heat source for the heat input point generated by the arc during welding, i.e. the heat generated during welding by the corresponding welding current and voltage. Therefore, when the heat input q and the thermal conductivity λ of the specimen are relatively constant, the temperature at a given point on the deposited specimen and the length of the distance from the heat source are inversely proportional. For single pass multilayer deposition specimens, the temperature at the bottom of the specimen is lower relative to the top when there is continuous deposition at the top. When the top is exothermic and the bottom is endothermic, the entire specimen tends to have a constant temperature and cool uniformly (Figure 4).

During the continuous deposition process, the specimen accumulates more heat because continuous deposition does not allow for cooling between layers. After ongoing heat accumulation, if deposition continues on the specimen, the molding quality of the deposited specimen deteriorates, the melting width at the top of the specimen increases, and the entire specimen will form a shape that is wide at the top and narrow at the bottom. The continuous high temperature in the specimen is equivalent to multiple heat treatments on the formed bottom area, which also changes the microstructure of the deposited specimen, resulting in degradation of the mechanical properties. Therefore, it is important to explore how to reduce the influence of the thermal history on the macromorphology, microstructure, and mechanical properties of specimens.

### 3.2. Process Design and Macromorphology

In view of the influence of the complex thermal history on the molding quality and microstructure of specimens in the WAAM process, we propose a method that involves cooling between deposition layers and reducing welding current (i.e. reducing heat input) with the increase of deposition layers to optimize the molding quality and microstructure of specimens. When the total number of deposition layers is 18, we determined that the four heat inputs and inter-layer cooling process parameters given in Table 3 are ideal.

Figure 5 shows the macromorphology of the cross sections of specimens deposited by different processes. Specimens A, C, and D had the same test duration, about 11 min. Specimen B had the longest processing time (about 60 min) because of the need for inter-layer cooling. Regarding the molding quality of Specimen A, the bottom forming width was narrow and the top was wide, resulting in a forming collapse. Figure 5b shows a deposition specimen for inter-layer cooling (Specimen B). The overall shape of the specimen was narrow, but the shape of each area was uniform. The height of Specimen B was slightly higher than that of Specimen A. Although the good molding quality of Specimen B was obtained by inter-layer cooling, the deposition rate was too slow. In order to improve the deposition rate and reduce deposition time, we tried gradually reducing the heat input. Figure 5c shows a deposition specimen in which the welding current for each layer was reduced by 2 A (Specimen C). The quality of the deposited samples obtained by this process was very poor, with a wide bottom and narrow top. Similar results were obtained by reducing by 1 A. Because the range of temperature change is larger at the beginning of the deposition and smaller at the later stage, it is not advisable to simply reduce the current, and no further study was made on Specimen C. Through a large number of tests, the formation process shown in Figure 5d was finally obtained. Every six layers were combined into a region, with a 3 A reduction for each layer of the first six layers, 2 A reduction for each layer of the middle six layers, and 1 A reduction for each two layers of the last six layers. As illustrated in the figure, the molding quality of the deposited section for Specimen D was better, with upper and lower widths of the specimen essentially the same.

According to the above results, the change of the temperature field in the deposition process leads to different shapes of the alloy deposit, especially when more heat is accumulated, as metal droplets then flow more freely. The quality of the alloy deposit can be optimized by inter-layer cooling and heat input control. However, inter-layer cooling requires a longer test time, and the heat input control needs to be adjusted according to the change in temperature and deposition height. In contrast, the deposition rate is faster and more practical when the heat input is adjusted for different regions.

Figure 6 shows side profiles of specimens deposited with different processes. Specimens A, B, and D have side molding with less roughness. The side roughness of Specimen D is the lowest, while that of Specimen B is the highest. Compared with the specimen formed without inter-layer cooling, the surface temperature of Specimen B was lower, which affected the melting width of the specimen. One can judge from the above information that adding inter-layer cooling makes the side molding of deposited specimens rough.

Because Specimens A and D had the same test time, we compared the temperature change of the specimens during the deposition process. Figure 7 shows the temperature curve of Specimens A and D during the deposition process, measured by placing thermocouples in the first layer. It can be seen that, the temperature of Specimen D was generally lower than that of Specimen A. During the deposition process, the specimens experienced a heating stage, a constant temperature stage, and a cooling stage. The alloy deposits and substrate could not rise to the maximum temperature in the first pass, but after several depositions, they could reach and maintain a relatively high temperature. Therefore, we determined that controlling the heat input can reduce the temperature of the specimen during the deposition process, thus reducing the heat effect on the alloy deposits.

### 3.3. Microstructure of Specimens with Different Processes

Figure 8 shows the microstructure of different areas from three processes observed under a 100× metallographic microscope. For Specimen A, there were columnar, planar, and dendritic crystals at the top, large columnar crystals in the middle, and equiaxed crystals at the bottom. The top of Specimen B was mostly dendrite, the middle was columnar, and the bottom was equiaxed. The top of Specimen D was dendrite with equiaxed grains, the middle was columnar and dendrite, and the bottom was equiaxed.

When the heat input was reduced layer by layer, the heat accumulation in the specimen was lower than that in the process with no change in heat input, and grain growth was inhibited at a lower temperature. In all three specimens, the microstructure of the upper part was larger than that of the lower part. When deposited at the bottom, the temperature of the alloy deposits was close to room temperature, and the crystallization and cooling time of metal droplets was faster. However, when deposition was carried out in the upper part of the specimen, which was at a higher temperature, the crystallization and cooling time of metal droplets became longer. These causes led to different grain shapes in different regions of the alloy deposits.

### 3.4. EBSD Analysis of Specimens with Different Processes

In order to study the effect of different deposition processes on grain formation, the Al-Mg alloy deposits were analyzed by EBSD, and the bottom region of the specimens with a built-up heat effect were observed (as shown in Figure 9). The grains were classified into recrystallized, substructured, and deformed, as divided by the grain boundary angle. It can be seen that Specimen B had the most recrystallized grains (see Figure 9b-2), and the specimen with the reduced heat input process had the least (see Figure 9c-2).

The results show that the average grain sizes produced by the three processes were similar, but the grain size distribution range of the specimens with inter-layer cooling was the largest, and that of the specimens with reduced heat input was the smallest. The inter-layer cooling process lasted longer and had a larger temperature range, and the longer and more complex thermal history resulted in an inhomogeneity of grain size in the alloy deposits. Recrystallization occurred in all three processes, which is due to the influence of gravity and temperature, and the grains grew along the direction of heat dissipation. The temperature of the specimen remains at the recrystallization temperature for a period of time when the specimen is affected by repeated heat during the deposition process, and the new grains grow up and gradually replace the original grains grown by thermal effect. The sequence of recrystallization involves the disappearance of adjacent subgrains and the combination of subgrains, and this process does not affect the crystal structure of grains [16,17]. Compared with the other two processes, the controlled heat input process (Specimen D) has a shorter test time and lower temperature, so it produces the fewest coarse recrystallized grains, along with a smaller average grain size and more uniform grain size. Therefore, changing the heat input process is beneficial to obtain a more uniform and fine microstructure.

We analyzed the texture of the deposited specimens with Channel 5 software (as shown in Figure 10). Interestingly, the two specimens without inter-layer cooling both had a rotated cube texture <110>, but the specimen of inter-layer cooling was Brass and part Cube texture.

During the deposition process, the grains grow along the direction of heat dissipation, and the stress state of the crystals also determines the orientation of the grains. When the inter-layer cooling process is used, the specimen experiences many more large-scale temperature fluctuations, and the stress state of the specimen is quite different from that of a specimen without inter-layer cooling, which leads to differences in texture and crystal growth direction. In addition, when the Al-Mg alloy continues to be deposited on the surface of the specimen after inter-layer cooling, the temperature is significantly lower than that during continuous deposition, another cause for different texture types. Thus, overall, different texture types are produced by inter-layer cooling. The results show that the grain texture strength and grain orientation of the alloy deposit generated by the inter-layer cooling process are higher, which leads to different properties in different directions. In conclusion, compared with the inter-layer cooling process, the process of controlled heat input is more suitable for manufacturing specimens with uniform microstructure.

### 3.5. Microhardness Analysis

The microhardness of the deposited specimens treated by different processes was tested. We tested from the bottom of the deposited specimens upward, and the distance between the two test points was 0.5 mm. Figure 11 shows the microhardness of the deposited specimen. The average microhardness of Specimen A was 71.38 HV; that of Specimen B was 76.59 HV; and that of Specimen D was 79.17 HV.

The microhardness of the inter-layer cooling specimen (Specimen B) fluctuated sharply due to the longer test time and larger temperature change. The fluctuation in microhardness of the two specimens without inter-layer cooling was relatively small. The average microhardness of the specimen with controlled heat input (Specimen D) was higher and the fluctuation of the value was the smallest. In addition, the microhardness of the bottom region of the three specimens was relatively low, because the bottom region had the longest thermal history. The thermal history also affected the microhardness, especially the fluctuation range of microhardness, which indirectly reflected the influence of microstructure on properties. It is evident that the process of controlled heat input can not only obtain better microstructure, but also improve microhardness.

## 4. Conclusions and Prospects

The molding quality of aluminum-magnesium ER5356 alloy was optimized by controlling inter-layer cooling and heat input reduction, and we analyzed the macromorphology, microstructure, and microhardness of the deposited specimens. The conclusions are as follows:(1)The results show that the specific heat capacity (C), thermal diffusion coefficient (α), and thermal conductivity (λ) of Al-Mg alloy deposit specimens do not change greatly with the increase in specimen temperature. The temperature of a certain point in the specimen during deposition is inversely proportional to the length of the distance from the heat source.(2)The process of inter-layer cooling and the process of controlled heat input both produce better molding quality. Specimens with reduced heat input have less sidewall roughness and higher deposition rate.(3)In the deposition process, a controlled heat input will make the overall temperature of the specimen lower than the process where the heat input is not changed, and the whole process is characterized by rising temperature, constant temperature, and cooling.(4)The average grain size of bottom equiaxed grains in the three processes is similar. However, the grain size distribution of the specimen from the inter-layer cooling process is the largest, and the grains are the most heterogeneous, with the most grown recrystallized grains. The specimen with controlled heat input has the most uniform, smallest average grain size, and the fewest larger recrystallization grains.(5)The bottom regions of the two specimens without interlaminar cooling both have a rotational cube texture <110>. The specimens from the inter-layer cooling process show Brass and part Cube texture. The longer processing time and the larger multiple temperature changes result in a more complex thermal history and more stress, which changes the texture type.(6)The specimens with regional control of heat input have the highest average microhardness, and the fluctuation of microhardness is the smallest. The bottom of the deposited specimen is affected by the longest thermal history, resulting in low microhardness.

We propose this process of controlling heat input by region because it results in a higher deposition rate, better molding quality, and better microstructure. In future work, we plan to study the number of zones, the formulation of heat input parameters, and changes in the mechanical properties from this process, so as to promote the application of arc additive manufacturing of Al-Mg alloy wire in engineering.

## Figures and Tables

**Figure 1 materials-14-01061-f001:**
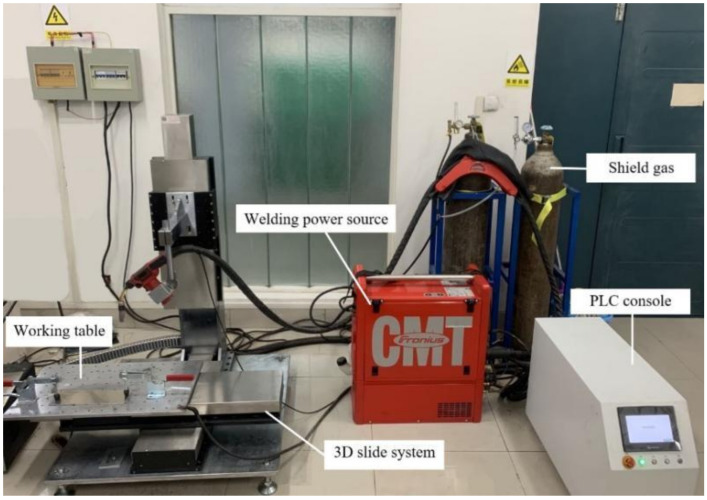
The cold metal transfer wire arc additive manufacturing (CMT-WAAM) system.

**Figure 2 materials-14-01061-f002:**
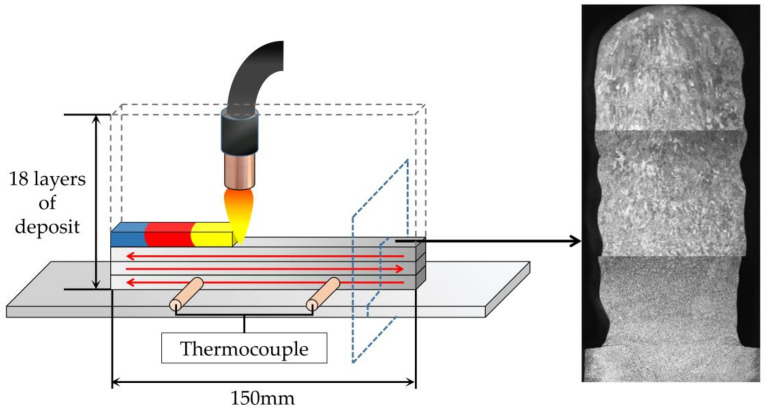
Schematic diagram of deposition process and specimen cutting.

**Figure 3 materials-14-01061-f003:**
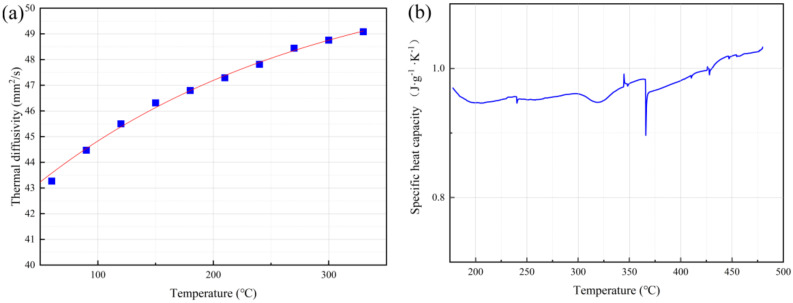
(**a**) Thermal diffusivity and (**b**) specific heat capacity of Al-Mg alloy deposited specimen.

**Figure 4 materials-14-01061-f004:**
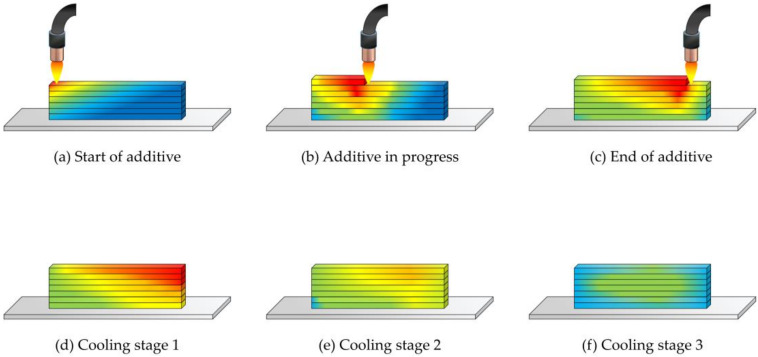
Variation of the temperature field in the process of wire arc additive manufacturing.

**Figure 5 materials-14-01061-f005:**
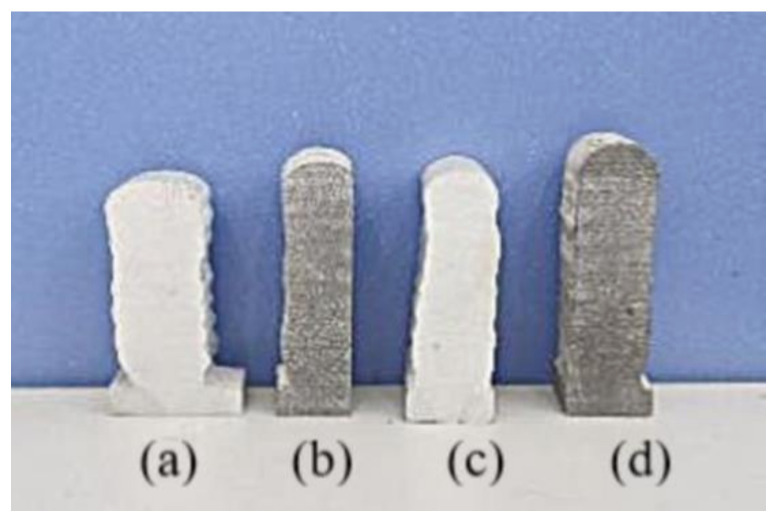
Macromorphology of deposited specimens with different processes. (**a**) Deposition specimen with constant heat input and inter-layer cooling (Specimen A). (**b**) Deposition specimen with inter-layer cooling (Specimen B). (**c**) Deposition specimen with welding current for each layer reduced by 2 A (Specimen C). (**d**) Specimen with proposed deposition process (Specimen D).

**Figure 6 materials-14-01061-f006:**
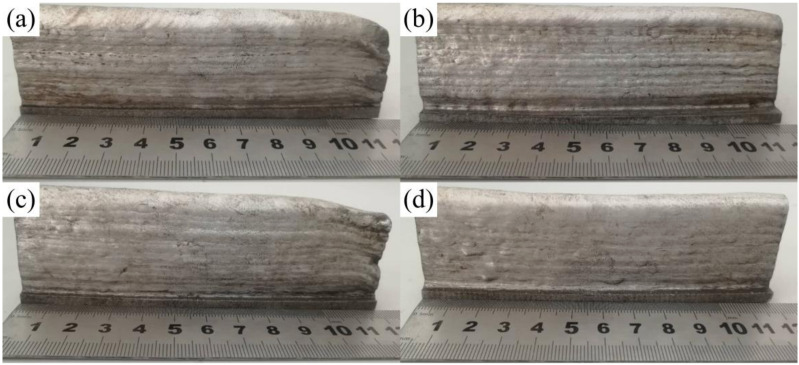
Macromorphology of deposited specimens with different processing. (**a**) Deposition specimen with constant heat input and inter-layer cooling (Specimen A). (**b**) Deposition specimen with inter-layer cooling (Specimen B). (**c**) Deposition specimen with welding current for each layer reduced by 2 A (Specimen C). (**d**) Specimen with proposed deposition process (Specimen D).

**Figure 7 materials-14-01061-f007:**
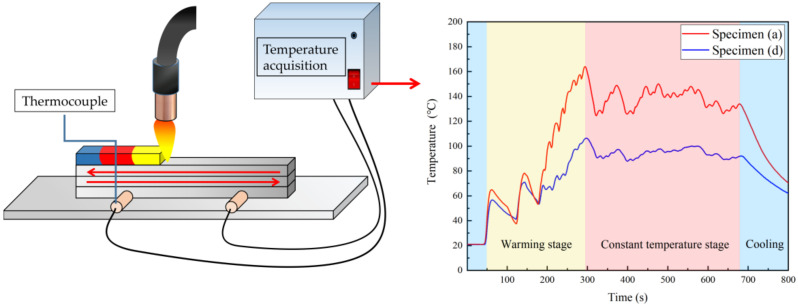
Temperature curve of specimens during WAAM.

**Figure 8 materials-14-01061-f008:**
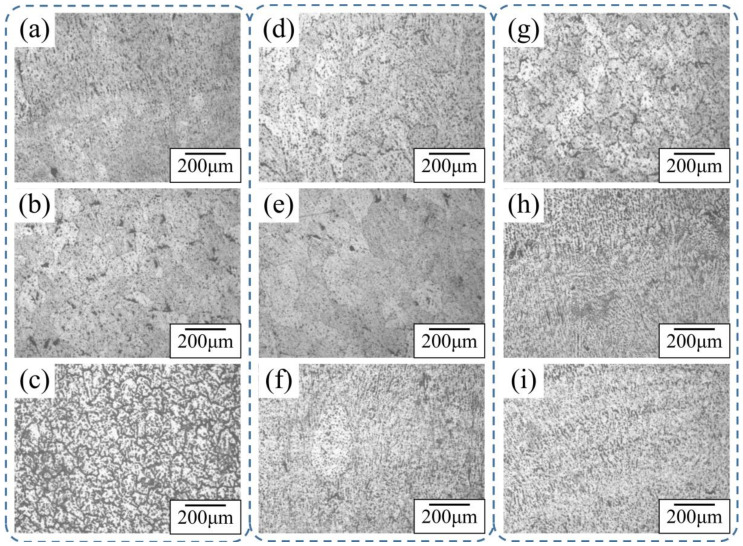
Microstructure of specimens from different processes. (**a**) Structure of top section of Specimen A. (**b**) Structure of middle section of Specimen A. (**c**) Structure of bottom section of Specimen A. (**d**) Structure of top section of Specimen B. (**e**) Structure of middle section of Specimen B. (**f**) Structure of bottom section of Specimen B. (**g**) Structure of top section of Specimen D. (**h**) Structure of middle section of Specimen D. (**i**) Structure of bottom section of Specimen D.

**Figure 9 materials-14-01061-f009:**
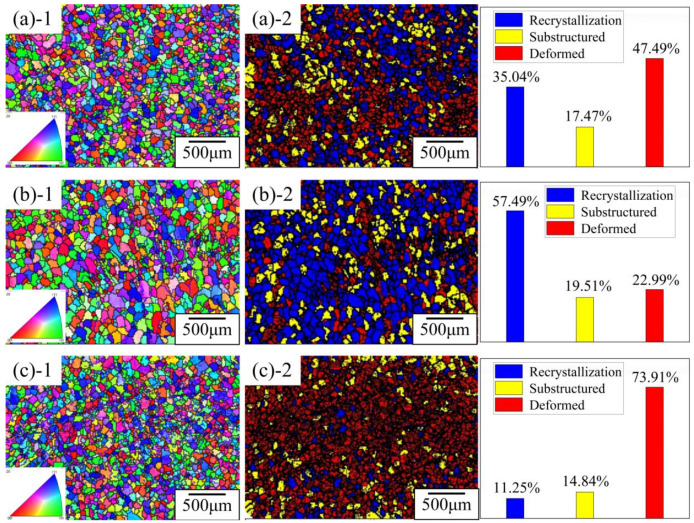
Electron back-scatter diffraction (EBSD) orientation maps and recrystallization diagram at the bottom of specimens from different processes. (**a**) No change of heat input and no cooling (Specimen A). The grain size is between 13.5 μm and 174 μm, and the average grain size is 47 μm. (**b**) No change of heat input, cooling between layers (Specimen B). The grain size ranges from 9.95 μm to 171 μm, and the average grain size is 48 μm. (**c**) Change of heat input without inter-layer cooling (Specimen D). The grain size ranges from 13.5 μm to 144 μm, and the average grain size is 43 μm.

**Figure 10 materials-14-01061-f010:**
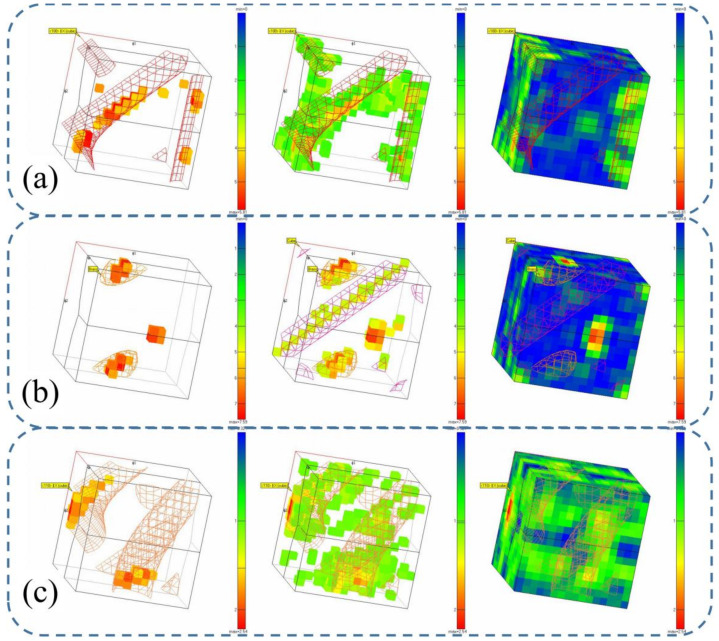
Optimal decision function (ODF) orientation map of specimens. (**a**) No change of heat input and no cooling (Specimen A). (**b**) No change of heat input, cooling between layers (Specimen B). (**c**) Change of heat input without inter-layer cooling (Specimen D). The grid is the region with strong texture strength for different kinds of textures; the regions with higher texture strength of alloy deposits treated by different processes are accurately distributed in the standard grid.

**Figure 11 materials-14-01061-f011:**
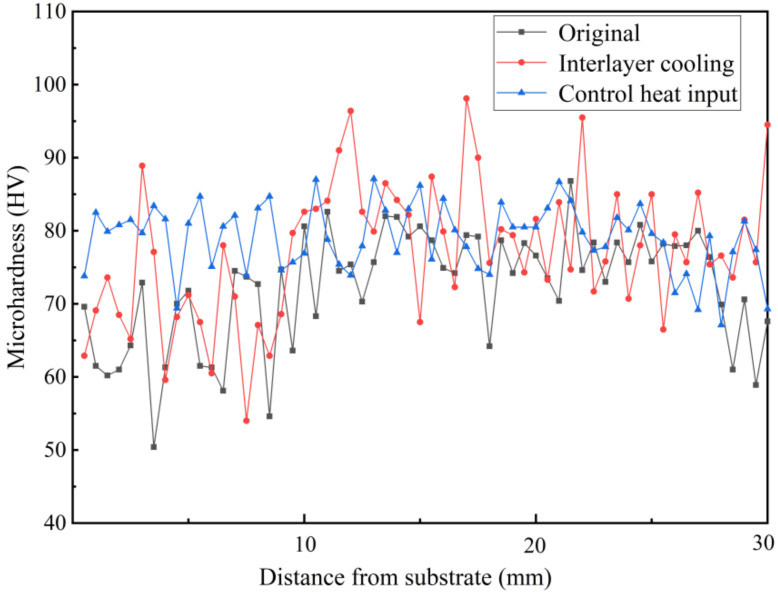
Microhardness of deposited specimens.

**Table 1 materials-14-01061-t001:** Chemical composition of ER5356 welding wire (mass fraction, %).

Alloys	Mg	Cr	Si	Fe	Cu	Zn	Mn	Ti	Al
ER5356	<5.5	<0.2	0.25	0.4	0.1	0.05	<0.2	<0.2	balance
6061	1.0	0.3	0.58	0.41	0.30	< 0.2	< 0.15	< 0.05	balance

**Table 2 materials-14-01061-t002:** Initial process parameters of Al-Mg alloy.

Process Type	Parameters
Current	150 A
Arc voltage	16.7 V
Travel speed	8 mm/s
Argon flow rate	20 L/min

**Table 3 materials-14-01061-t003:** Process parameter table for changing heat input and inter-layer cooling.

Code	Welding Heat Input	Inter-Layer Cooling System
a	Welding current 150 A	No cooling
b	Welding current 150 A	Inter-layer cooling for 3 min
c	Initial current 150 A Each layer decreases by 2 A	No cooling
d	Initial current 150 A The first six layers reduce 3 A by layer The middle six layers reduce 2 A by layer The next six layers reduce 1 A by 2 layers	No cooling

## Data Availability

The data presented in this study are available on request from the corresponding author.

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
