# Peer review of "Regional Control and Optimization of Heat Input during CMT by Wire Arc Additive Manufacturing: Modeling and Microstructure Effects"

_materials, 2021, doi:10.3390/ma14051061_

Round 1
Reviewer 1 Report
In this study, the authors develop and describe a method for optimizing regional control on the heat input for wire arc AM by CMT. The work presented is interesting, if somewhat lacking in originality. The figures and layout are acceptable, but the quality of the language needs a significant amount of improvement. I also have some concerns with the presentation and the technical aspects of the study. However, I think that they can be adequately addressed by the authors without redoing the study completely. Therefore, I recommend the paper be sent back to the authors for a major revision before another round of review. My detailed comments are below.
Presentation-Related Comments
1. As previously stated, the quality of English usage in this paper is poor. Some sentences are almost unreadable and do not support the generally sound technical aspects of the work. Large sections of the paper seem like they were simply machine translated (e.g., using Google Translate) into English without much polishing or clean-up afterwards. I strongly recommend consulting a language service during the revision of the paper.
2. The title is confusing and doesn't make sense when the abstract is read. It should be something like "Regional Control and Optimization of Heat Input During CMT by Wire Arc Additive Manufacturing: Modeling and Microstructure Effects"
3. The abstract needs to be improved significantly. Based on it alone, it seems the authors simply made and tested some samples without a clear "why" or explanation of the results. This might be acceptable in a technical note when the data collected is useful on its own, but not for a full research article. Much more is actually presented in the paper and the abstract should reflect this.
4. The introduction section is very poor. This is almost no literature review and very little description of the problem and what has been done to address it already. I would expect to see at least one additional page for this section in the revised version of the paper.
5. At the end of Section 1, there is a technical description of the study but little to no discussion of the novelty and originality of the study. What is offered by this work that has not been done in other work?
6. The authors (at least the corresponding author) must use academic or professional email addresses. Using personal email services like gmail, hotmail, qq.com, and 123.com is not acceptable. Please update.
7. Figure 9: Please use a clearer/higher resolution version of this figure in the final version of the paper
8. There is no solid, useful discussion section in the paper. There are a lot of interesting results and they are not discussed much. This needs to be improved in the final version of the paper. I would expect to seen at least 2-3 additional pages of discussion in the revised paper.
Technical Comments
9. Table 1: Was this data actually collected by the authors or taken from a datasheet? If collected, give details. If taken from a datasheet, cite the datasheet.
10. Figure 2: Please update the figure to show the location of the thermocouple and conductivity measuring tool during processing
11. Figure 4: This is an interesting figure but could be misleading. One of the most important considerations with metal AM (wire arc or powder) is the residual stresses that form in the printed structure. This diagram implies that the structure cools at a fairly even rate from bottom to top, which is not usually the case unless the printing was done inside a furnace or done very fast (far, far faster than the 8mm/s the authors used). Cooling will happen quickly behind the arc and it will be uneven, with top layer cooling the fastest. It would be best to make sure your diagram accurately portrays this.
12. Figure 10 is interesting but not discussed in enough detail. Please greatly expand the discussion here and its impact on AM modeling and process design.
Author Response
Dear reviewer,
The comments have been carefully took into account and a new revised submission have been uploaded. The modified part has been marked.The responses are as follows,
To question 1: Through consulting the relevant language institutions, the English of the whole article has been reorganized and improved.
To question 2: At the beginning, the title of the article really bothered me. Thank you for your comments and suggestions on the title of this article. The title has been changed to "Regional Control and Optimization of Heat Input During CMT by Wire Arc Additive Manufacturing: Modeling and Microstructure Effects".
To question 3: The abstract section of the manuscript has been reorganized and edited.
To question 4: The introduction of the article has been revised according to the suggestions. Clear "why" for this study and related tests.
To question 5: The introduction of the article has been modified according to the modification suggestions. The related references are reorganized and explained, and the existing problems and solutions are described.
To question 6: At the end of Section 1, a description of the novelty and originality of the study is added.
To question 7: The email address has been modified.
To question 8: In the new revised draft, many new discussion sections have been added. It includes the analysis of forming quality of alloy deposits under different processes, the further discussion of microstructure, the difference of recrystallized grains under different processes, and the introduction and discussion of different texture types caused by different processes.
To question 9: The data in Table 1 were collected, including the chemical composition of ER5356 welding wire produced by ESAB company and 6061 aluminum alloy substrate produced by China national standard. I only found the documents provided by the relevant companies, but failed to find the source of the corresponding references. The relevant data are as follows:
To question 10: The thermocouple is shown in the corresponding position in Figure 2. The equipment for measuring the thermal conductivity is a separate equipment, which needs to sample the deposited specimens and test the thermal conductivity at different temperatures. Therefore, the location of the device cannot be shown in Figure 2.
To question 11: The point you put forward is something we haven't considered before. The temperature change diagram of alloy deposit during additive manufacturing has been rearranged. Thank you for your comments.
To question 12: The discussion of Figure 10 has been added.
We appreciate for reviewer’s warm work earnestly, and hope that the correction will meet with approval.
Once again, thank you very much for your comments and suggestions.

Reviewer 2 Report
The manuscript is well-organized, and in principle the subject fits within the scope of 'materials'. The authors used wire arc additive manufacturing process to manufacture 5356 aluminum alloy parts using cold metal transfer methods. The English and references are satisfactory but the manuscript presents some parts to be revised.
1. The notation for the aluminum alloy must be revised through the paper according to journal format.
2. Line 72, it is beneficial to the reader if the wall length of 150mm also indicated in Figure 2.
3. Line 96, "change of thermal diffusivity of 5356 aluminum alloy has no obvious effect on the heat conduction phenomenon"
→ AM process is very local and heat sensitive. What is the reason for this judgment? Any references?
4. In figure 5, what is the total process time for each specimen?
5. Fig. 8 and Fig. 9 are illegible. Please clarify the font and Figures. Maybe enlarge the figure 8 and 9.
6. The authors had an detailed discussion on the results of microstructural analysis. But how does this affect mechanical properties of the AM parts?
Author Response
Dear reviewer,
The comments have been carefully took into account and a new revised submission have been uploaded. The modified part has been marked. The responses are as follows,
To question 1: The notation of aluminum alloy in the manuscript has been modified as required.
To question 2: In Figure 2, the marking of 150 mm wall length of alloy coating has been updated in the figure.
To question 3: In the first draft, I'm sorry that I didn't describe this problem clearly. What I want to describe is that the specific heat capacity, thermal diffusivity and thermal conductivity of the alloy deposit will not fluctuate sharply with the change of temperature after measuring the thermophysical properties of the alloy deposit by relevant instruments. So, these parameters of the material will not change dramatically due to the change of temperature, and most of the factors affecting the heat conduction are the distance from the heat source. The introduction and description of this problem have been reorganized. Thank you for your comments.
To question 4: Specimens (a), (c) and (d) have the same test duration, about 11 min. Specimen (b) needs more time, about 60min, because it needs interlayer cooling. I've added these parameters to Section 3.2.
To question 5: Figures 8 and 9 have been modified and enlarged.
To question 6: For this study, in the previous test, we only carried out the microhardness test. Microhardness analysis of alloy deposits under different processes has been added to section 3.5.
We appreciate for reviewer’s warm work earnestly, and hope that the correction will meet with approval.
Once again, thank you very much for your comments and suggestions.

Reviewer 3 Report
Dear authors,
Thank you for the article on the important area of additive manufacturing. Below you will find my comments.
1) It seems inevitable to develop an introduction on WAAM and aluminum alloys. How does it look for this or other series aluminum alloys (not necessarily using the CMT method)?
2) Little to no discussion in the Results and Discussion section.
3) How was the chemical composition analysed (row 62)? Is the result in weight or atomic percent?
4) What technique was used for cutting (row 73)? WEDM?
5) Table 3 is difficult to understand - it proposes to remodel it into a matrix.
6) Magnification over 100x is not worth highlighting (row 191).
7) Which is 0.1% in Figure 9c-3?
8) I consider the lack of any mechanical tests to be a big disadvantage of the article (even the results of hardness measurements).
Other editorial notes: no space before the unit (rows 56 and 60), parameters (row 63) open and unclosed parentheses (row 206), unify the term for macroscopic examination: macromorphology vs. macro morphology vs. macroscopic morphologies (rows 134, 144 and 145).
Author Response
Dear reviewer,
The comments have been carefully took into account and a new revised submission have been uploaded. The modified part has been marked. The responses are as follows,
To question 1: More descriptions and introductions of WAAM have been added in Section 1.
To question 2: In the new revised draft, many new discussion sections have been added. It includes the analysis of forming quality of alloy deposits under different processes, the further discussion of microstructure, the difference of recrystallized grains under different processes, and the introduction and discussion of different texture types caused by different processes.
To question 3: The data in Table 1 were collected, including the chemical composition of ER5356 welding wire produced by ESAB company and 6061 aluminum alloy substrate produced by China national standard. I only found the documents provided by the relevant companies, but failed to find the source of the corresponding references. The original file can be seen in the attachment. The relevant data are as follows:
To question 4: As for the cutting technology of the specimen, I can't explain it clearly in the first draft. This problem has been revised in the new revised draft.
To question 5: Table 3 has been remade in matrix format.
To question 6: Figure 8 has been remade and rearranged. Thank you for your advice.
To question 7: In the first draft, the 99% ratio of figure 9c-3 is the test error, including the noise of EBSD image. In order to cause misunderstanding and trouble, I have redrawn Figure 9. Thank you for your careful review.
To question 8: For this study, in the previous test, we only carried out the microhardness test. Microhardness analysis of alloy deposits under different processes has been added to section 3.5.
We appreciate for reviewer’s warm work earnestly, and hope that the correction will meet with approval.
Once again, thank you very much for your comments and suggestions.

Round 2
Reviewer 1 Report
Dear Authors,
Thank you for the revisions to the manuscript. I am satisfied with the paper as it is now and recommend acceptance and publication.
Reviewer 3 Report
I have no comments